

# Application of a new scheme of cloud base droplet nucleation in a Spectral (bin) Microphysics cloud model: sensitivity to aerosol size distribution

**E. Ilotoviz and A. Khain**

**Department of Atmospheric Sciences, The Hebrew University of Jerusalem, Israel**

Submitted to Atmos. Chem. Phys. Discuss.

June 2016

Revised

21 September 2016

Communicating author: Alexander Khain, Department of Atmospheric Sciences, The Hebrew University of Jerusalem, Israel, email: alexander.khain@mail.huji.ac.il

**Abstract**
A new scheme of droplet nucleation at cloud base is implemented into the Hebrew University
Cloud Model (HUCM) with spectral (bin) microphysics. In this scheme, supersaturation
maximum $S_{max}$ near cloud base is calculated using theoretical results according to which
$S_{max} \sim w^{3/4} N_d^{-1/2}$ , where $w$ is the vertical velocity at cloud base and $N_d$ is droplet concentration.
Microphysical cloud structure obtained in the simulations of a mid-latitude hail storm using the
new scheme is compared with that obtained in the standard approach, in which droplet nucleation
is calculated using supersaturation calculated in grid points. The simulations were performed
with different concentrations of cloud condensational nuclei (CCN) and with different shapes of
CCN size spectra. It is shown that the new nucleation scheme substantially improves the vertical
profile of droplet concentration shifting the concentration maximum to cloud base. It is shown
that the effect of the CCN size distribution shape on cloud microphysics is not less important
than the effect of the total CCN concentration. It is shown that the smallest CCN with diameters
less than about 0.015 $\mu m$ have a substantial effect on mixed-phase and ice microphysics of deep
convective clouds. Such CCN are not measured by standard CCN probes which hinders
understanding of cold microphysical processes.
Key words: cloud-aerosol interaction, droplet nucleation at cloud base, spectral bin
microphysics

## 1. Introduction

Droplet concentration is the key microphysical parameter that affects precipitation formation, and radiative cloud properties (Pruppacher and Klett, 1997). The droplet concentration determines major microphysical cloud properties such as height of precipitation onset, type of precipitation (liquid, mixed phase and ice) (Khain, 2009; Freud and Rosenfeld, 2012; Tao et al. 2012) . Droplet concentration is determined by concentration and size distribution of aerosol particles (AP) and by the maximum value of supersaturation near cloud base $S_{max}$. $S_{max}$ is reached at a few tens of meters above cloud base (Rogers and Yau, 1996). The vertical grid spacing of most cloud-resolving models is too coarse to resolve this maximum. This can lead to errors in determination of droplet concentration. Therefore, it is desirable to parameterize the process of droplet nucleation near cloud base. One approach to the parameterization is based on lookup tables developed using precise 1D parcel models (e.g., Segal and Khain, 2006). The other approach is based on analytical calculation of supersaturation maximum, $S_{max}$, near cloud base. This approach has been developed in several studies using various assumptions concerning CCN activity spectra (Ghan et al., 1993, 1997; Bedos et al., 1996; Abdul-Razzak et al., 1998; Cohard et al., 1998; Abdul-Razzak and Ghan, 2000; Fountoukis, 2005; Shipway and Abel, 2010). In these studies calculation of a supersaturation maximum is reduced to solving a complicated integro-differential equation assuming different expressions for CCN activation spectra. The parameters of activation CCN spectra, as well as the concentration and shape of the CCN size distributions, are often prescribed in atmospheric models and assumed to be invariant over time. The results and a comparison of these approaches are presented by Ghan et al. (2011).

In cloud models with a comparatively high resolution (Kogan 2001; Khain et al. 2014)
supersaturation $S_w$ is calculated explicitly at each grid point. In these bin microphysics models AP
playing the role of cloud condensational nuclei (CCN) are described using aerosol size distribution
functions containing several tens of size bins. The value of supersaturation is used to calculate the
critical radius of AP using the Köhler theory. All CCN with sizes exceeding this critical value are
activated to droplets.    This approach will be referred to as standard approach (ST) where
supersaturation maximum near cloud base is not resolved and the vertical profile of
supersaturation may not contain such maximum. It leads to underestimation of droplet
concentration in clouds, at least in  their low part.
In set of studies by Pinsky et al. (2012, 2013, 2014) formation of profiles of supersaturation
and of droplet concentration were investigated both analytically and by means of a high precision
model of an ascending adiabatic parcel. Pinsky et al. (2012) proposed a simple method of
calculating  $S_{max}$ near cloud base for monodisperse aerosol size distribution. The detailed test
showed that the method can be applied to any CCN spectra. Pinsky et al (2014) gave a theoretical
basis for such conclusion by calculating droplet concentrations using multidisperse size spectra of
AP. The method of calculating droplet concentration near cloud base using $S_{max}$ will be referred to
as *new approach* (NA).
In this study we investigate the effects of application of NA on the microphysics of mid-
latitude deep convective clouds (hail storm) using the Hebrew University Cloud model (HUCM)
with spectral-bin microphysics (SBM).   The effect of the new approach is investigated in
simulations with different parameters of CCN activation spectra.

**2. Model description**
The HUCM is a 2-D, nonhydrostatic SBM model with  microphysics based on solving a
system of equations for size distributions of liquid drops, three types of pristine ice crystals
(plates, columns, and dendrites), snow/aggregates, graupel, hail and partially frozen or "freezing
drops". Each size distribution is discretized into 43 mass-doubling bins, with the smallest bin
equivalent to the mass of a liquid droplet of radius 2 $\mu m$. AP playing the role of CCN  are also
defined on a mass grid containing 43 mass bins. The size of dry CCNs ranges from 0.005 $\mu m$ to 2
$\mu m$.
Primary nucleation of each ice crystal type is described using Meyers et al. [1992]
parameterization. The type of ice crystals is determined depending on temperature range  where
the particles arise (Takahashi et al., 1991). Secondary ice generation is taken into account during
riming (Hallett and Mossop 1974).  Collisions are described by solving the stochastic collection
equations for the corresponding size distributions using the Bott (1998) method. Height-
dependent, gravitational collision  kernels for drop-drop and drop-graupel interactions are taken
from Pinsky et al. (2001) and Khain et al. (2001); those for collisions between ice crystals are
taken from Khain and Sednev (1995) and Khain et al. (2004). The latter studies include the
dependence of particle mass on the ice crystal cross-section. The effects of turbulence on
collisions between cloud drops are included (Benmoshe et al. 2012). The collision kernels depend
on the turbulence intensity and changes over time and space.
The time-dependent melting of snow, graupel, and hail as well as shedding of water from hail
follows the approach suggested by Phillips et al. (2007). We have implemented liquid water mass
in these hydrometeor particles that is advected and settle similarly to the mass of the
corresponding particles. As a result, these particles are characterized by their total mass and by the
mass of liquid water (i.e., the liquid water mass fraction). The liquid water fraction increases
during melting. As soon as it exceeds ~95%, the melting particles are converted to raindrops.
Process of time dependent freezing is described according to Phillips et al. (2014, 2015). The
freezing process consists of two stages. The first nucleation stage is described using the
parameterization of immersion drop freezing proposed by Vali (1994) and Bigg (1953). Drops
with radii below 80 $\mu m$ that freeze are assigned to plates, whereas larger drops undergoing
freezing are assigned to freezing drops. The freezing drops consist of a core of liquid water
surrounded by an ice envelope. Time-dependent freezing of liquid within freezing drops is
calculated by solving the heat balance equations that take into account the effects of accretion of
supercooled drops and ice particles. Collision between freezing drops and other hydrometeors lead
either to the freezing drops category if the freezing drop is larger than its counterpart. Otherwise,
the resulting particle is assigned to the type of counterpart. Once the liquid water fraction in a
freezing drop becomes less than some minimal value (<1%) it is converted to a hailstone. Hail can
grow either by dry growth or by wet growth (Phillips et al. 2014, 2015). Accordingly, liquid water
is allowed in hail and graupel particles at both positive and negative temperatures. The shedding
of water in wet growth is also included.
Water accreted onto aggregates (snow) freezes immediately at temperatures below $0^0 C$,
where it then contributes to the rimed fraction. This rimed mass distribution is advected and settle
similarly to the snow masses. Riming mass increases the density of the aggregates. As the bulk
density of snow in a certain mass bin exceeds a critical value (0.2 $g\ cm^{-3}$), the snow from this bin
is converted into graupel. The appearance of water on the  surface of hailstones as well as an
increase in the rimed fraction of snowflakes affect the particle fall velocities and coalescence
efficiencies.
The initial size distribution of CCN (at t=0) is calculated using the empirical dependence (i.e.,
the Twomey formula) of concentration $N_{ccn}$ of activated CCN on supersaturation $S_w$ (in %)
$N_{ccn} = N_o S_w^k$, where $N_o$ and $k$ are the measured constants (Khain et al., 2000). The obtained
aerosol size distribution is corrected in zones of very small and very large CCN, that is, in size
ranges where the Twomey formula is invalid.    At t>0 the prognostic equation for the size
distribution of non-activated CCN is solved. Using the value of $S$ calculated at each time-step and
in each grid point, the critical radius of CCN particles was determined according to the Köhler
theory. The CCNs with radii exceeding the critical value are activated and new droplets are
nucleated. The corresponding bins of the CCN size distributions become empty. In ST, this
procedure is used at all cloud grid points.
In  NA, droplet concentration at cloud base is calculated using the formula for $S_{max}$  derived
by
Pinsky et al. (2012)

$$S_{max} = C w^{3/4} N_d^{-1/2} ,\qquad\qquad (1)$$

where $w$ is vertical velocity at cloud base, $N_d$ is droplet concentration and  coefficient $C$  slightly
depends on the thermodynamical parameters only (see **Table 1** for notations). A brief derivation
of the formula (1) is presented in **Appendix**. Since the droplet concentration at cloud base is equal
to the concentration of CCN activated at $S_w = S_{max}$, the droplet concentration at the cloud base can
be calculated as:

$$N_d = \int_{r_{n\_cr}(S_{max})}^{\infty} f(r_n)\,dr_n \qquad\qquad (2)$$

where $f(r_n)$ is a size distribution of dry AP and $r_{n\_cr}$ is critical radius of CCN activated under
$S_{max}$. According to the Köhler theory, the critical radius relates to $S_{max}$ as
$$r_{n\_cr} = \frac{A}{3}\left(\frac{4}{BS_{max}^2}\right)^{1/3},\qquad(3)$$

where coefficients $A$ and $B$ are the coefficients of the Köhler equation for equilibrium
supersaturation (see Table 1 for notations). Substituting Eq. (2) into (1) one can obtain equation
for $S_{max}$:
$$S_{max}\underbrace{\left[\int_{r_{n\_cr}(S_{max})}^{\infty} f(r_n)dr_n\right]}_{N_d}^{1/2} = Cw^{3/4}\qquad(4)$$

Taking into account the relationship (3), Eq. (4) contains only one unknown $S_{max}$. This
equation is easily solved by iteration calculating $S_{max}$, $r_{n\_cr}(S_{max})$ and concentration of nucleated
droplets at cloud base at each time step.
The values of $S_{max}$ were calculated at all grid points corresponding to cloud base, which is
determined as the first grid point from below, at which $S_w \geq 0$.

## 3. Design of simulations

All simulations were performed within a computational domain of 153.9 km x 19.2 km, and
a grid spacing of 300 m in the horizontal direction and 100 m in the vertical direction. Effects of
NA on cloud microphysics were tested in simulations of a thunderstorm observed in Villingen-
Schwenningen, southwest Germany, on June 28, 2006. Meteorological conditions (including
sounding) of this storm are described by Khain et al. [2011]. The background wind direction was
quasi-2-D, which simplified the prescription of the background wind profile in the 2-D model. The
wind speed increased with height from ~10 $m\ s^{-1}$ in the lower atmosphere to about 20 $m\ s^{-1}$ at
levels of 100-200 mb. Surface temperature was 22.9℃, the relative humidity near the ground was
high (~85%), which led to a low lifting condensation level of about 890 m. The freezing level was
located at around 3.5 km. The observed maximum diameter of hailstones was about 5 cm.

The convection was triggered by a cool pool, which is typical in simulations of long-lasting

convection (Rotunno and Klemp, 1985).
Three sets of simulations were performed, each simulation in two versions: according to ST
where the critical CCN radius was calculated using a supersaturation calculated at the grid points
using the values of temperature and humidity, and according to NA where the critical CCN radius
and $S_{max}$ were determined from Eq. (9).

*The first set of simulations* aims at the comparison of the microphysics between NA and ST

in cases of high  ($N_0 = 3500\ cm^{-3}$)  and  low ($N_0 = 100\ cm^{-3}$) CCN concentrations. Minimum
CCN radii were set equal to 0.015 $\mu m$ and 0.0125 $\mu m$, respectively. These values correspond to
the data according to which the nuclei mode (the smallest CCN) in Marine aerosol size
distribution contains aerosols smaller than the nuclei mode in Continental case or even than in
Urban case (Ghan et al, 2011). Similar CCN size distributions were used by Khain et al (2011).
These simulations are referred to as E3500, E100 (T) and EN3500, EN100 (NA), respectively.
In *the second set of simulations* the smallest CCN were added into the AP spectra. The large
impact of the smallest CCN in formation of ice crystals in cloud anvils was shown by Khain et al.
(2012). The minimum CCN radii were taken equal to 0.006 $\mu m$ and 0.003 $\mu m$ in cases of high
and low CCN concentrations, respectively. These simulations are referred to as E3500-S, EN3500-
S, E100-S and EN100-S, where symbol "S" denotes small AP.
In the first and the second sets of simulations the slope parameter $k$ was assumed equal to 0.9.
*The third set of simulations* was similar to the second one, but with the slope parameter $k$
=0.5.  In many studies investigating effects of aerosols on cloud microphysics only parameter $N_0$
is changed. However, the slope parameter determines the relationship between concentration of
smaller and larger CCN, so concentration of nucleated droplets also depends on the slope
parameter.  The simulations of the third set are referred to as E3500-S-05, EN3500-S-05, E100-S-
05 and EN100-S-05. Size distributions of CCN in the  simulations are shown in **Figure 1**.
CCN concentrations in  the simulations s are presented in **Table 2**. Although the difference
between the total aerosol concentrations is not large, in case k=0.5 the CCN size distribution
contains more large CCN and fewer small CCN. These size distributions were assumed within the
lower 2-km layer. Above this level, the CCN concentration in each mass bin was decreased
exponentially with height. Above 8 km, the CCN concentration was set constant.

**4. Results of simulations**

**4.1  Vertical profiles of supersaturation near cloud base**
The model  calculates supersaturation at the  model grid points which typically do not
exactly coincide with the cloud base level where supersaturation $S_w$=0. We consider the first level
where $S_w \geq 0$ as the cloud base. Since the supersaturation maximum is reached not far from the
cloud base level, especially for high AP concentration cases (Pinsky et al. 2012), we attribute the
values of $S_{max}$ to this level. Correspondingly, the difference between NA and ST in the droplet
concentrations is also attributed to this level. **Figure 2** shows vertical profiles of supersaturation
calculated in ST and NA  simulations in the atmospheric columns where the velocity at cloud base
was equal to 1 $ms^{-1}$.   It is natural that the values of $S_{max}$ are larger in case of low CCN
concentration as compared to high CCN concentration case. For goals of the present study, a more
interesting finding is that the values of $S_{max}$ calculated using NA are substantially larger than $S_w$
calculated at model level associated to the cloud base in ST. The difference between NA and ST in
the supersaturation values leads to a substantial difference in the droplet concentrations, especially
in cases of high CCN concentration. Calculation of $S_{max}$ at cloud base changes the vertical profile
of supersaturation above it. While in ST supersaturation changes only slightly or even increase
with height within  100-200 m above cloud base, in NA supersaturation decreases within this layer
above the supersaturation maximum in agreement with the theory (Rogers and Yau, 1989, Pinsky
et al, 2012, 2013).

To justify the values of supersaturation and droplet concentration obtained in NA,

benchmark simulations using a parcel model were performed. The parcel model describes AP and
drops using drop size distribution defined on a mass grid containing  2000 mass bins (Pinsky et al,
2002). It calculates growth of AP and droplets by solving the equation for diffusional growth
written in the most general form without using parameterization of droplet nucleation. Time step
used for solving the diffusional growth equation was 0.001 s. The model was used earlier for
developing lookup tables relating parameters of AP and vertical velocity to droplet concentration
(Segal and Khain, 2006). Simulations with the parcel model were performed for the same vertical
velocity at cloud base, temperature and CCN distributions as in the HUCM simulations. As can be
seen from Fig. 2, the values of supersaturation and droplet concentration calculated using NA are
much closer to those calculated using the parcel model as compared to  the values calculated using
ST.
The model level associated with the cloud base (where $S_w \geq 0$ ) is slightly higher than the
lifting condensation level (LCL), where $S_w = 0$. At the same time, the calculations performed
according to Pinsky et al. (2012) show that the level where $S_w = S_{max}$ is located  from about 20 m
(for high CCN concentration) to about 60 m (for low CCM concentration) higher than the LCL.
The estimations show, therefore, that the level where $S_w = S_{max}$ is quite close to the model cloud
base level. Accordingly, the droplet concentration determined at $S_w = S_{max}$ is assigned to the
corresponding grid point at the model cloud base.

*4.2  High CCN concentration*
In this section we compare the results for three pairs of simulations of clouds were
developing in a highly polluted atmosphere. **Figure 3** shows the fields of droplet concentration $N_d$
at the developing stage of the cloud evolution in E3500-S-0.5 (a), EN3500-S-0.5 (b), E3500-S (c)
and EN3500-S (d). The maximum $N_d$ in NA is reached at cloud base, which makes the cloud base
well pronounced. The difference between droplet concentrations in ST and NA experiments
decreases with height. The highest droplet concentration is reached in simulations where the CCN
activation spectrum was characterized by the slope parameter k=0.5 . This can be attributed to the
fact that at  k=0.5 the aerosol spectrum contains more CCN which are activated at cloud base than
at k=0.9.
Vertical profiles of the maximum values of droplet concentration and of cloud water content
(CWC) averaged over time periods of storm development (a-b) and over the mature stage (c,d) are
presented in **Figure 4.**

In NA the $N_d$ maximum is reached near cloud base and the droplet concentration decreases

with height.. This behavior of $N_d(z)$ is more realistic than in ST, where $N_d$ increases with height
up to an altitudes 2- 4 km, depending on the stage of storm evolution. This increase in the $N_d$ in
ST is caused by in-cloud activation of mid-size CCN which were not activated at cloud base in the
standard approach. In NA, these CCN were activated at cloud base. There is, therefore, a negative
feedback in the supersaturation-droplet concentration relationship: an underestimation of
supersaturation at low levels in the ST simulations leads to the underestimation of droplet
concentration and to the corresponding increase in supersaturation at comparatively small
distances above cloud base. These results indicate that in models where droplet nucleation is
calculated only at cloud base, the correct calculation of $S_{max}$ at cloud base is *strictly necessary to*
*obtain reasonable values of* $N_d$ in clouds.

At height of about 4-5 kms, droplet concentrations in ST and NA become nearly similar.

Figs. 4a,c show also that $N_d$ is very sensitive to the slope parameter of the CCN activation
spectrum. The maximum $N_d$ reached at cloud base is about $1100\,cm^{-3}$ in EN3500-S-05 (k=0.5)
as compared to $\sim 550\,cm^{-3}$ in EN3500-S (k=0.9). This difference is caused by the fact that in case
k=0.5 the concentration of CCN with sizes exceeding $\sim 0.015\,\mu m$ (which are activated at cloud
base) is larger than in case k=0.9 (see Fig.1).

The effect of the *smallest* CCN on $N_d$ (and on entire ice microphysical structure) becomes

very important above 6 km. In simulations containing the smallest CCN, these CCN are activated
producing new small droplets at heights of around 6.5- 8 km. The increase in $N_d$ is shown in Fig.
4a,c by red arrows. These smallest CCN are not activated at cloud base even in NA (where $S_{max}$
is larger than $S_w$ in ST). This in-cloud nucleation is caused by an increase in supersaturation at
these levels due to a decrease in CWC (Fig. 4b,d) and an increase in vertical velocity (not shown).
The increase in $N_d$ by activation at high levels and its effect on concentration of ice crystals in
cloud anvils of deep convective clouds was also reported by Khain et al. (2012).
Since the slope parameter determines concentration both of larger CCN and of smallest
CCN, the slope parameter also affects the concentration of droplets nucleated at high levels.
Vertical profiles of CWC (Figs. 4b,d) are typical of deep convective clouds developing in
the highly polluted environment: CWC is large and has maximum at about 5 km, i.e. at quite high
altitude.
**Figure 5a** shows the vertical profiles of maximum concentration of plate crystals (in HUCM
homogeneous freezing leads to formation of plates) averaged over the mature stage of cloud
evolution (from 4860 to 5460s).  The number concentration of ice crystals in E3500 and EN3500
(in which there are no the smallest CCN in the initial CCN spectrum) is by factor of 5 lower than
in simulations with the CCN spectra containing the smallest CCN. The results show that ice
crystal concentration in NA is higher only slightly than in ST. Thus *the concentration of ice*
*crystals in cloud anvils is determined to a large extent by  the concentration of smallest CCN in*
*the CCN spectra and is substantially less sensitive to larger CCN, which are activated at cloud*
*base*. Figure 5b shows that this conclusion is valid for the entire period of the simulation.  In
agreement with Fig. 4c, the concentration of plates increased when NA was used (**Fig. 5b**). The
comparative contribution of the smallest CCN and CCN additionally activated at the cloud base in
NA (as compared to ST) are shown in Fig. 5b by arrows.
**Figure 6** shows the vertical profiles of time averaged maximum mass contents of  ice
crystals, snow, graupel and hail+freezing drops at the storm mature stage. The maximum
difference between ice crystal mass contents takes place at ~10-11 km, where ice crystals are
caused by homogeneous freezing.

The most pronounced effect of NA is an increase in the accretion rate. In agreement with

results of simulations of aerosol effects on ice microstructure of deep convective clouds (Khain
2009; Tao et al. 2012; Khain et al. 2016), the intensification of riming leads to a decrease in the
snow mass content and to an increase in the mass contents of graupel (Fig.6b-c).. The existence of
the smallest CCN concentration leads to further decrease in the snow mass content and to the
increase in the graupel mass content. This smallest CCN lead to higher supercooled droplet
concentration and to an increase in the liquid mass available for riming (Fig. 4d,e).

## 312 4.3. Low CCN concentration

In this section we compare the results for three pairs of simulations: a) E100 and EN100, b)

E100-S and EN100-S, and c) E100-S-0.5 and EN100-S-0.5 in which clouds were developed in the
atmosphere with low CCN concentration. After the first 35 min of cloud evolution, the cloud base
is located at 700-800 m altitude and T=16.8℃ at this level.

The fields of droplet concentration $N_d$ in different simulations at the developing stage of the

cloud evolution are shown in **Figure 7**. The maximum $N_d$ in a NA is reached at cloud base, which
makes the cloud base well pronounced. The difference in droplet concentrations between ST and
NA simulations decreases with height**.** Although the difference is $N_d$ between NA and ST is very
pronounced, the absolute difference is not large (about 20 $cm^{-3}$). This low $N_d$ determines a
typical maritime microphysical structure of clouds in both NA and ST cases.
**Figure 8** shows vertical profiles of the maximum values of droplet concentration and cloud
water content (CWC) averaged over the time period of 3420-4020s (mature stage). One can see a
dramatic difference in the profiles of droplet concentration and between CWC values of  at low
CCN concentration as compared to high CCN concentration ( Fig. 4). At  low CCN concentration,
droplet collisions are efficient and droplet concentration decreases with height much faster than in
polluted air. As a result, the CWC maximum  at  low CCN concentration is located at the height of
2 km as compared to 5 km in case of high CCN concentration. These  differences determine the
huge difference in the ice microphysics.
Fig. 8 shows that both the droplet concentration and CWC are larger in NA as compared to
ST. The main differences between droplet concentrations near cloud base are, however,
determined by the difference in the slope parameter value:  at  k=0.5 there are more CCN of sizes
exceeding 0.015 $\mu m$  than   at k=0.9 (Fig. 1).These CCN are activated at cloud base  leading to
higher concentration in simulations with k=0.5, especially when NA was applied.
Efficient collisions (seen by the sharp decrease in the CWC above z=2 km)  and rain fall
decrease the droplet concentration. As a result, the supersaturation increases and leads to in-cloud
nucleation and an  increase in the droplet concentration already at distances of a few hundred
meters above the cloud base. However, since the concentration of CCN is  low,  the amount of
new nucleated droplets in the simulations was only about 5-10 $cm^{-3}$ . The second layer of intense
in-cloud nucleation caused by activation of the smallest CCN is seen within the altitude layer from
4 km to 8 km. The difference in droplet concentration within this layer is fully related to the
existence/absence of smallest CCN in the CCN size spectrum. The differences between droplet
concentration in ST and NA simulations are not significant at these levels.
This result agrees with  the case of high CCN concentration when  droplet concentration at
higher levels is to a large extent determined by the smallest CCN in the droplet spectrum.
**Figure 9** presents the vertical profiles of maximum mass contents of ice crystals, snow,
graupel and hail + freezing drops at the mature stage of cloud evolution. Comparison  with Fig. 6
shows that with the exception of snow, the mass contents of different ice hydrometeors at low
CCN concentration are substantially lower than at high CCN concentration. The main reason for
such difference is lower CWC at low CCN concentration that leads to less intense riming and,
consequently to slow growth of ice particles.
Fig. 9 shows that the profiles of ice hydrometeors in NA and ST are similar. It means that
the ice microphysics is to a large extent determined by the mass of supercooled droplets at high
levels which in turn is determined by the *smallest* CCN in the CCN size spectrum. The effects of
the smallest CCN and the shape of CCN size spectra on droplet concentration and the
concentration on ice microphysics are much stronger than the effect of additional droplets
nucleating at cloud base in the NA. The reason for this effect was explained above.
The increase in the concentration of the smallest CCN and in droplet concentration leads to
an increase in the ice crystals mass content occurring about the level of homogeneous freezing
(Fig.9a).
The mass content of snow decreases with the increase in the smallest CCN concentration ,
because intensification of riming of snow leads to its conversion to graupel (Fig. 9b).
Consequently, the graupel mass content increases (Fig. 9c).  As regards to mass content of hail,
the increase in the smallest CCN concentration leads to a decrease in the hail content above 6 km
and to its increase below this level (Fig. 9d).  The higher  hail mass content above 6 km layer in
the absence of  smallest CCN is likely related to the fact the low droplet concentration leads to
formation of raindrops in high concentration. Although these raindrops are of comparatively small
size, the total raindrop mass content is larger than that in case of higher drop concentration. These
raindrops rapidly freeze above the freezing level producing hail (actually frozen drops) with total
mass larger than at high CCN concentration. This effect is discussed by Ilotovich et al. (2016) in
detail. In HUCM, frozen raindrops are assigned to the hail category due to their high density. If
hail is defined as particles with sizes exceeding 1 cm, the amount of hail at low CCN
concentration would be negligible.

Higher hail mass content below 6 km in the presence of the smallest CCN can be attributed
to intense conversion of heavy rimed graupel to hail, as well as to more efficient hail growth by
riming. Note that sizes of hail particles forming in a deep convective cloud developing in the
polluted atmosphere are larger than hail forming in a cloud developing in clean air (Ilotovich et al.
2016). Due to larger size, hail in the polluted case falls to the surface (Fig. 6d), while in clean air
hail melts at 1.5 km in the absence of small CCN, and in vicinity of the surface if the CCN size
spectrum contains the smallest CCN.
**4.3 The impact on precipitation**
**Figure 10a** shows the accumulated rain at surface in the polluted air . Accumulated rain
is maximum in EN3500-S-0.5 where effect of smallest CCNs is combined with the effect of
comparatively large amount of large CCN. This synergetic effect of the smallest and large CCN
is described by Khain et al. (2011). In most simulations, the masses of accumulated rain are
quite similar.
Comparison of Fig. 10a and Fig. 10 b shows that the accumulated rain at low aerosol
concentration is lower than at high CCN concentration, which is in agreement with many
previous studies. Accumulated rain in NA was found to be quite close to that in ST. The main
difference in the values of accumulated rain at low CCN concentration is caused by effects of
smallest aerosols increasing the mass of precipitating ice particles .
Amount of hail at the surface in polluted air (**Figure 10c**) is substantially larger than in
clean air (**Figure 10d**) due to lower sizes and faster melting of hail particles if CCN
concentration is low. The effect of AP on the size and amount of hail at the surface was
investigated by Ilotovich et al. (2016) in detail.
Amount of hail at the surface in polluted air is slightly higher in EN3500-S-0.5 as
compared to E3500-S-0.5 (**Figure 10c**). We attribute this effect to a higher rate of riming in
EN3500-S-0.5 due to a higher amount of supercoold water (Fig. 4b, d). There are no
significant differences in the other cases of polluted air.
The main factor determining the differences in the amount of hail falling to the surface at
low CCN concentration is the effect of smallest CCN. The increase in concentration of smallest
CCN leads to an increase in hail growth by riming.
As regards to the ratio of hail amounts in the experiments with smallest AP, earlier or later
intensification of convective cells (which is more or less random) may affect the ratio. Since
the mass of hail falling to the surface in clean air is very low, a larger computational area is
required to obtain reliable statistics.

**5. Conclusions**
Sensitivity of the microphysics of deep convective clouds to the concentration of aerosols and
to the shape of aerosol size distribution is investigated using a new version of a 2D Spectral (bin)
Microphysics Cloud Model (HUCM). A new component of the model is the calculation of
maximum supersaturation at cloud base using the analytical expression derived by  Pinsky et al.
(2012). The cloud microphysical structure obtained using this expression is compared with that
obtained with supersaturation calculated  at  model grid points.
The goal of the study was twofold: a) to test the  effects of the improved calculation of
supersaturation maximum near cloud base (NA (new approach)  vs  ST  (standard approach)) at
different aerosol loadings and b) to evaluate sensitivity of cloud microphysics to concentration and
shape of size distribution of aerosol particles. In the simulations, shape of CCN size distributions
was changed by changing the value of the slope parameter in the expression for activation
spectrum (the values of k=0.5 and k=0.8 were used) and by adding the smallest CCN with radii
below 0.015 $\mu m$.
The values of $S_{max}$ near cloud base calculated by the theoretical analysis were found  to be
substantially larger than the supersaturation values calculated explicitly at model grid points
associated with cloud base. The comparison of the values of supresaturation at cloud base and
droplet concentration in the model simulations with the corresponding values calculated using a
benchmark parcel model showed that NA simulates cloud base supersaturation and droplet
concentration much more accurately than ST.  Thus, *the first main conclusion* of the study is that
the droplet concentration field in NA is substantially more realistic than in ST, with the maximum
of droplet concentration in NA located near cloud base in agreement with classical results (Rogers
and Yau, 1989). The increased droplet concentration makes the cloud base more pronounced. The
improvement of the representation of the vertical profile of the droplet concentration is especially
significant in case of high CCN concentration, where utilization of $S_{max}$  leads to a substantial
increase in the concentration of droplets near cloud base. Thus, even at 100-m vertical resolution,
it is necessary to use analytical expressions for $S_{max}$ . At low CCN concentration, the improved
representation of droplet concentration above cloud base has a comparatively weak effect on cloud
microphysics. This result can be attributed to the fact that droplet concentration increases
relatively slightly if it is more accurately calculated since the available CCN concentration is low.
As a result, intense warm rain rapidly arises in both NA and ST.
The error in calculation of droplet concentration near cloud base in ST is compensated to a
significant extent by in-cloud nucleation above cloud base. Indeed, in NA droplet concentration
increases with height up to level of 4 km (Fig. 4a). The only reason of such increase is the in-cloud
nucleation of comparatively large CCN.
Models with microphysical schemes that do not describe in-cloud droplet nucleation should
include calculation of $S_{max}$ at cloud base to avoid large errors in simulation of the microphysical
cloud structure.
*The second main conclusion* is high importance of the shape of CCN size distribution. Cloud
microphysics was found to be highly sensitive to the slope parameter of the CCN activation
spectra. The effect is comparable with the change in the total CCN concentration via the change
in the intercept parameter $N_0$. The utilization of k=0.5 instead of k=0.9 nearly doubled droplet
concentration near cloud base that leads to corresponding effects on cloud microphysics, in
particular, to an increase in accumulated rain.
*The third main conclusion* is high sensitivity of ice microphysics to the existence of the
smallest CCN in the CCN size spectrum. Both in cases of low and high CCN concentration, the
differences in ice microphysics are determined to a large extent by *concentration of the smallest*
*aerosols in the CCN spectra*. In cases of high CCN concentration, the effect of the smallest CCN
in the NA becomes important above 5-6 km altitude where they are activated producing additional
supercooled liquid droplets. The latter leads to the increase in the concentration of ice crystals
above the level of homogeneous freezing by factor of about 5, to doubling of graupel mass
maximum. The smallest CCN also influence hail size and mass content.
In case of low CCN concentration the smallest CCN also lead to an increase in the
concentration and mass contents of ice crystals and to a significant increase of graupel and hail
mass contents. Note that many probes of CCN measure concentration of CCN at supersaturations
not exceeding 0.6%. In this case the concentration of the smallest CCN which remain non-
activated at this supersaturation remains unknown. Such measurements do not provide necessary
information for investigation of mixed-phase and ice microphysics.
Accumulated rain amount in case of high CCN concentration turned out to be higher than in
case of low CCN concentration. This result was discussed by Khain (2009) and Ilotovich et
al.(2016) showing that formation of hail increases precipitation efficiency of mid-latitude storms.
Ice precipitation (calculated in mm of melted hail) at the surface is much lower than liquid
precipitation. Nevertheless, hail precipitation at the surface in case of high CCN concentration is
higher than in case of low CCN concentration by order of magnitude in agreement with results by
Khain et al. (2011) and Ilotoviz et al. (2016). This effect can be attributed by formation of larger
hail particles in case of high CCN concentration (high supercooled mass content). The large hail
particles reach the surface, while smaller hail forming in case of low CCN concentration melts
without reaching the surface.
The concentrations of drops and ice crystals are important parameters determining cloud
radiative properties. In this context, more accurate calculation of the concentrations using the NA
as well as taking into account the effects of smallest CCN should improve the accuracy of
evaluation of radiative cloud properties.  The proposed approach of calculation of nucleation of
droplets at cloud base is simple in the utilization and computationally efficient. It can be used in
cloud-resolved models with different vertical grid spacing. The utilization of cruder vertical model
resolution may lead to larger errors in cases when droplet concentration at cloud base is calculated
using supersaturations calculated at model grid points.
*Acknowledgements*
The study is supported by the US Department of Energy Award DE_FOA-0000647 from the U.S.
Department of Energy Atmospheric System Research Program,  by the  Binational US-Israel
Science Foundation (grant 2010446), and by the Israel Science Foundation (grant 1393/14).

**Appendix. Derivation of an expression for the supersaturation maximum at cloud base**
Detailed description of the derivation of Eq. (1) is given in Pinsky et al. (2012). Below we present
only a short description.  Assuming that near cloud base $S_w \ll 1$, the equation for supersaturation
can be written as:

$$\frac{dS_w}{dt} = A_1 \frac{dz}{dt} - A_2 \frac{dq_w}{dt} \qquad\qquad\qquad (A1)$$

where coefficients $A_1$ and $A_2$ are presented in **Table 1**;  z is the height above cloud base and  $q_w$
is liquid water mixing ratio. The first term on the right-hand side of eq. (A1) describes an
increase in supersaturation due to adiabatic air cooling during ascent, whereas the second term
describes the supersaturation decrease caused by condensation of water vapor on droplets.
Integration of equation (A1) leads to the equation of mass balance:
$$S_w = A_1 z - A_2 q_w + C_1 \qquad \text{(A2)}$$
where $C_1 = 0$ at cloud base. Assuming monodisperse DSD with droplets of radii r, the liquid
water mixing ratio can be written as :
$$q_w = \frac{4}{3} \pi \frac{\rho_w}{\rho_a} N_d r^3 \qquad \text{(A3)}$$
where $N_d$ is the droplet concentration. The equation for diffusional growth can be written is the
form where the curvature term and the chemical term are omitted (Pinsky et al. 2012):
$$\frac{dr}{dt} = \frac{1}{Fr} S_w \qquad \text{(A4)}$$
The expression for coefficient F is presented in **Table 1**. Coefficients $A_1$, $A_2$ and F slightly
depend on temperature and can be assumed constant in the analysis. Using Eqs. (A2-A4), eq. (1)
can be rewritten in the closed form as:
$$\frac{dS_w}{dz} = A_1 - \frac{1}{w} B_1 (A_2 N_d)^{2/3} (A_1 z - S_w)^{1/3} S_w \qquad \text{(A5)}$$
where $B_1 = \frac{3}{F} \left( \frac{4\pi}{3} \frac{\rho_w}{\rho_a} \right)^{2/3}$
Pinsky et al. (2012) showed that Eq. (A5) can be written in a non-dimensional form that
results in an universal profile of supersaturation with height at given vertical velocity. The
condition $\frac{dS_w}{dz} = 0$ applied to this equation allows to get solution in the form (1) for $S_{max}$, as
well as for the height of $S_{max}$ over the cloud base. Pinsky et al. (2012, 2014) showed that (1) is
valid for any size distributions of CCN.

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

**Table 1. List of symbols**



| Symbol | Description | Units |
|---|---|---|
| $A$ | $\dfrac{2\sigma_w}{\rho_w R_v T}$ | m |
| $A_1$ | $\dfrac{g}{R_a T}\left(\dfrac{L_w R_a}{c_p R_v T}-1\right)$ | $m^{-1}$ |
| $A_2$ | $\dfrac{1}{q_v}+\dfrac{L_w^2}{c_p R_v T^2}$ | - |
| $B$ | $\dfrac{v_n \Phi_s \varepsilon_m M_w \rho_n}{M_n \rho_w}$ | - |
| $B_1$ | $\dfrac{3}{F}\left(\dfrac{4\pi\rho_w}{3\rho_a}\right)^{2/3}$ | $m^2 s$ |
| $C_1$ | $1.058(FA_1/3)^{3/4}\left(\dfrac{3\rho_a}{4\pi\rho_w A_2}\right)^{1/2}$ | $m^{9/4}\,s^{-3/4}$ |
| $c_p$ | specific heat capacity of moist air at constant pressure | $J\,kg^{-1}K^{-1}$ |
| $D$ | coefficient of water vapor diffusion in the air | $m^2\,s^{-1}$ |
| $e$ | | |
| $e_w$ | saturation vapor pressure above the flat surface of water | $N\,m^{-2}$ |
| $g$ | acceleration of gravity | $m\,s^{-2}$ |
| $F$ | $\left(\dfrac{\rho_w L_w^2}{k_a R_v T^2}+\dfrac{\rho_w R_v T}{e_w(T)D}\right)$ | $m^{-2}\,s$ |
| $K$ | parameter of activity spectra | |
| $k_a$ | coefficient of air heat conductivity | $J\,m^{-1}s^{-1}K^{-1}$ |
| $L_w$ | latent heat for liquid water | $J\,kg^{-1}$ |
| $M_n$ | molecular weight of aerosol salt | $kg\,mol^{-1}$ |
| $M_w$ | molecular weight of water | $kg\,mol^{-1}$ |
| $N_d$ | concentration of liquid droplets | $m^{-3}$ |

| | | |
|---|---|---|
| $N_0$ | parameter of activation spectra | |
| $P$ | pressure of moist air | N m$^{-2}$ |
| $q_v$ | water vapor mixing ratio air) | $kg\ kg^{-1}$ |
| $q_w$ | liquid water mixing ratio | $kg\ kg^{-1}$ |
| $r_{max}$ | drop radius at $z = z_{max}$ | m |
| | | - |
| $S_w$ | $S_w = e/e_w - 1$    supersaturation over water | - |
| $S_{max}$ | supersaturation maximum | - |
| $T$ | absolute temperature | $^{o}$K |
| $T_C$ | temperature at cloud base | $^{o}$C |
| $w$ | vertical velocity | m s$^{-1}$ |
| $z$ | height over condensation level | m |
| $z_{max}$ | height of supersaturation maximum | m |
| | | |
| $\varepsilon_m$ | soluble fraction | - |
| $\rho_a$ | density of air | kg m$^{-3}$ |
| $\rho_N$ | density of a dry aerosol particle | kg m$^{-3}$ |
| $\rho_w$ | density of liquid water | kg m$^{-3}$ |
| $\sigma_w$ | surface tension of  water-air interface | Nm$^{-1}$ |
| | | |
| $\nu_n$ | van 't Hoff factor | - |








**Table 2**. CCN concentrations in different experiments in the boundary layer

| | High CCN concentration, $cm^{-3}$ | | Low CCN concentration, $cm^{-3}$ | |
|---|---|---|---|---|
| Slope parameter | No smallest CCN | With smallest CCN | No smallest CCN | With smallest CCN |
| k=0.9 | 840 | 2930 | 33 | 214 |
| k=0.5 | 1552 | 3140 | 53 | 152 |
















# Figures


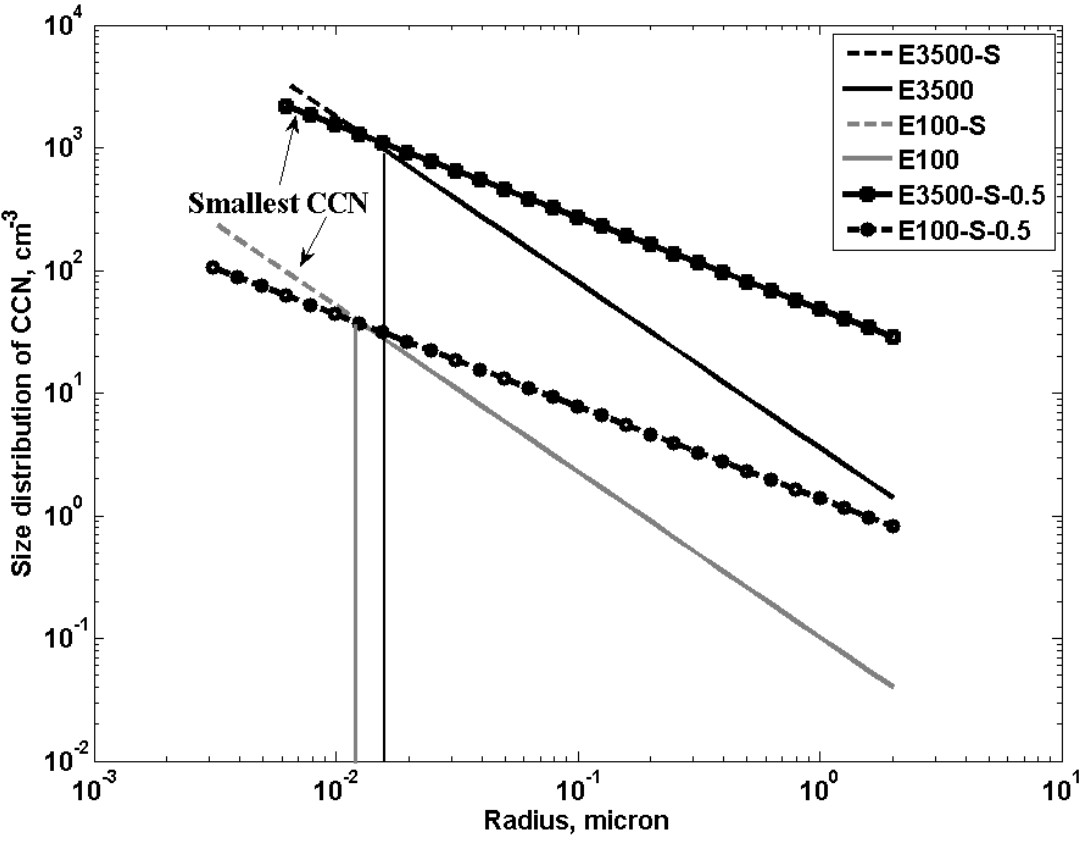


**Figure 1.** The initial size distributions of aerosols near the surface in different simulations.


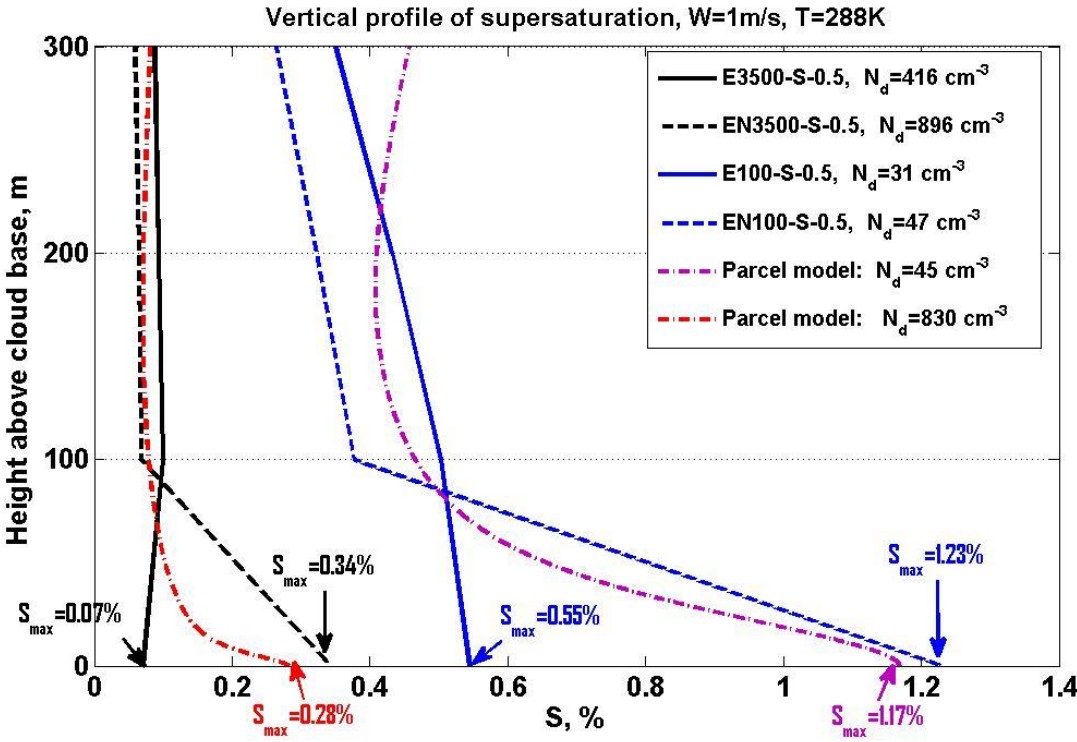


**Figure 2.** Examples of vertical profiles of the supersaturation above cloud base calculated using HUCM and a benchmark parcel model. The columns with w close to 1 m/s at cloud base were chosen for comparison. The values of $S_{max}$ in HUCM were calculated according to *Pinsky et al.* (2012). The values of droplets concentration calculated at cloud base in different simulations are shown as well (see legend box).










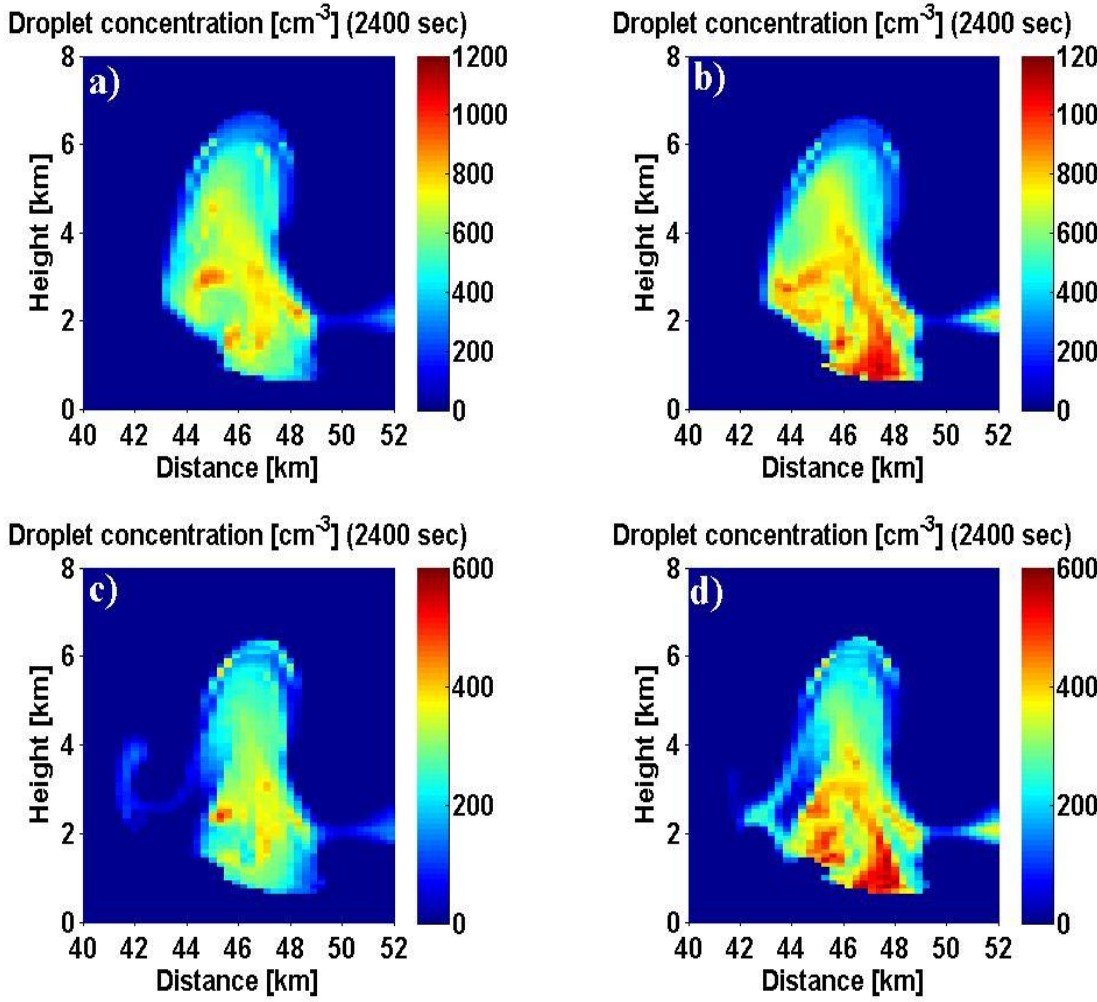




**Figure 3.** Field of droplet concentration at t=2400s in (a) E3500-S-0.5, (b) EN3500-S-0.5, (c) E3500-S and (d) EN3500-S.





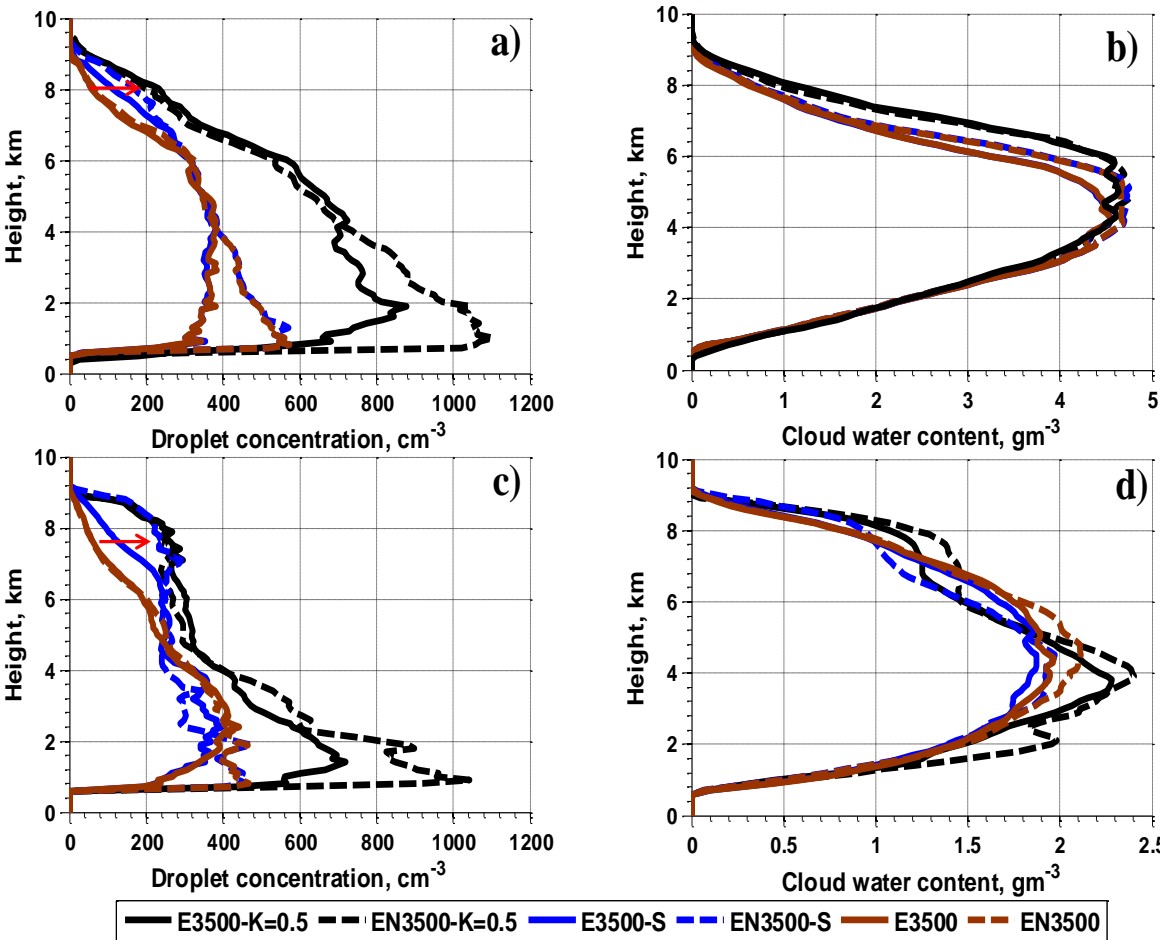



**Figure 4.** Vertical profiles of the maximum values of droplet concentration (a,d) and CWC(b,e) in simulations with high CCN concentration. The profiles are obtained by averaging over the time period of 2400-3000s (upper row) and over time period of 4860-5460s (bottom row). Panel (c) shows a zoom of panel (b) for large CWC .



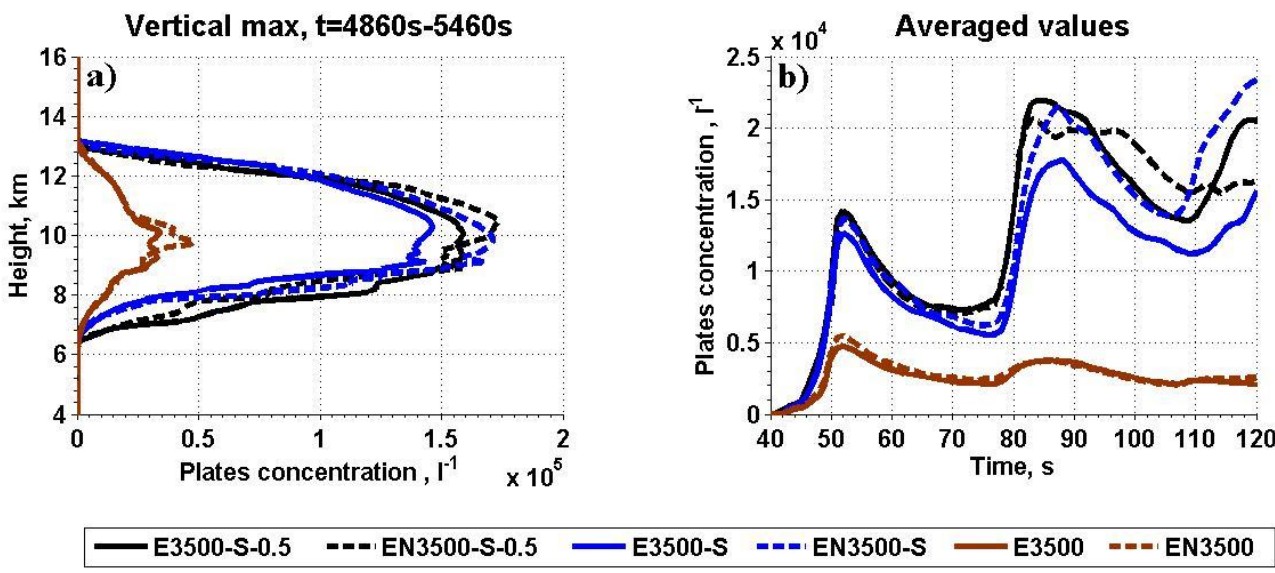





**Figure 5.** Vertical profiles of (a) maximum values of plates concentration and (b) time dependencies of averaged plate concentration. The profiles are obtained by averaging over the time period of 4860-5460s. The low and the upper arrows in the panel b show approximate contribution of smallest CCN and the additional CCN activated in NA, respectively.






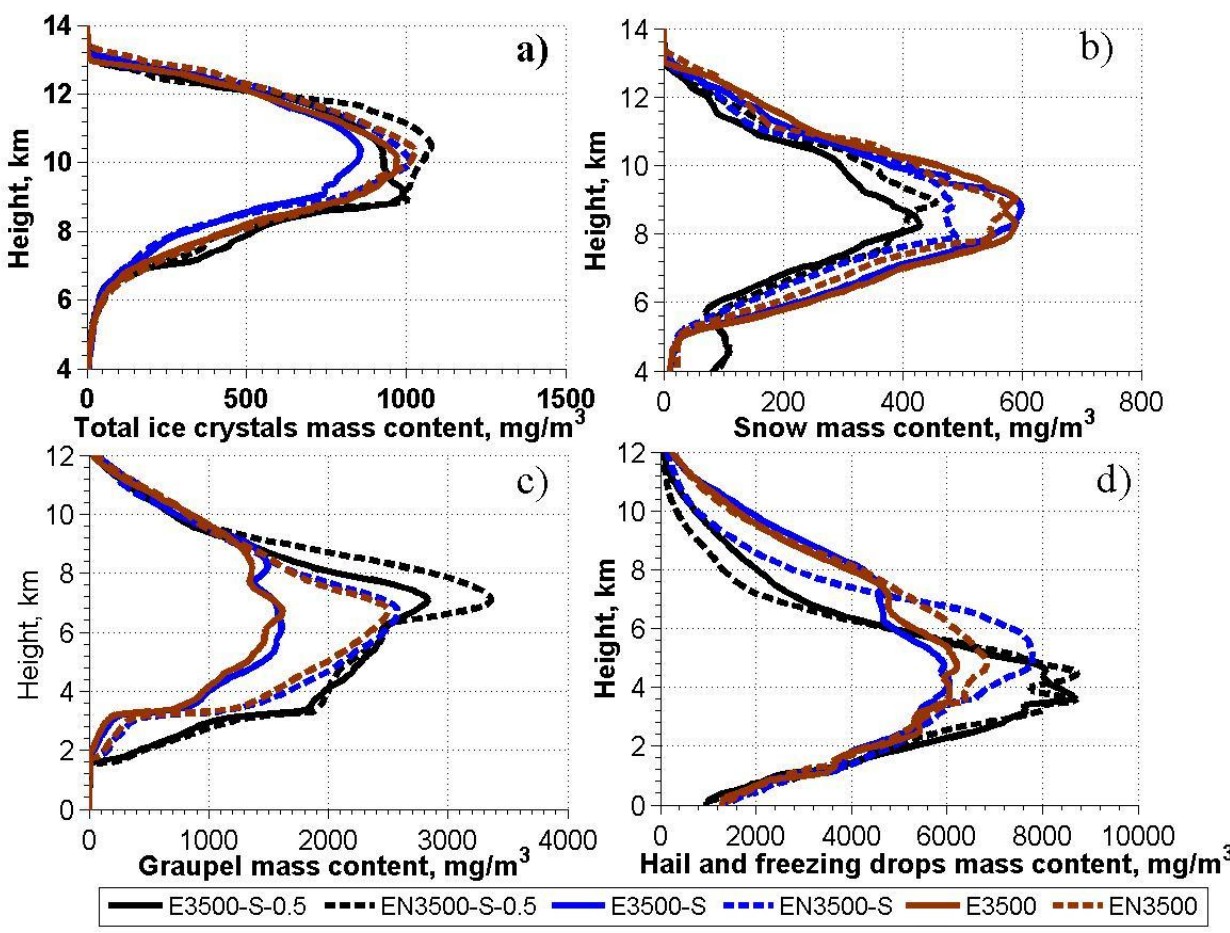



**Figure 6.** Vertical profiles of the maximum values of mass content: (a) total ice crystals, (b) snow, (c) graupel and (d) total hail and freezing drops in simulations with high CCN concentration. The profiles are obtained by averaging over the time period of 4860-5460s.












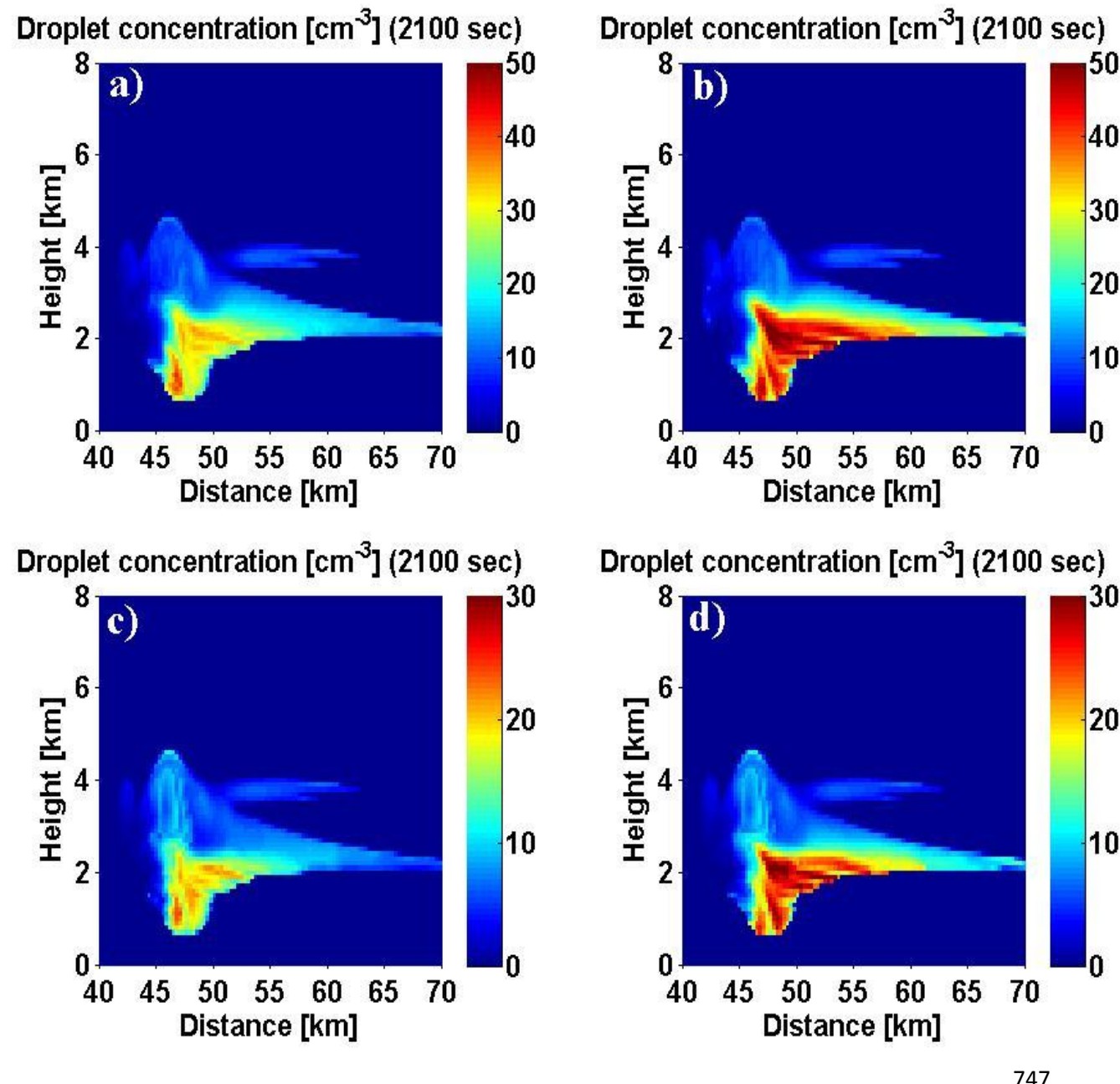




**Figure 7.** Field of droplet concentration at t=2100s in (a) E100-S-0.5, (b) EN100-S-0.5, (c) E100-S and (d) EN100-S simulations.






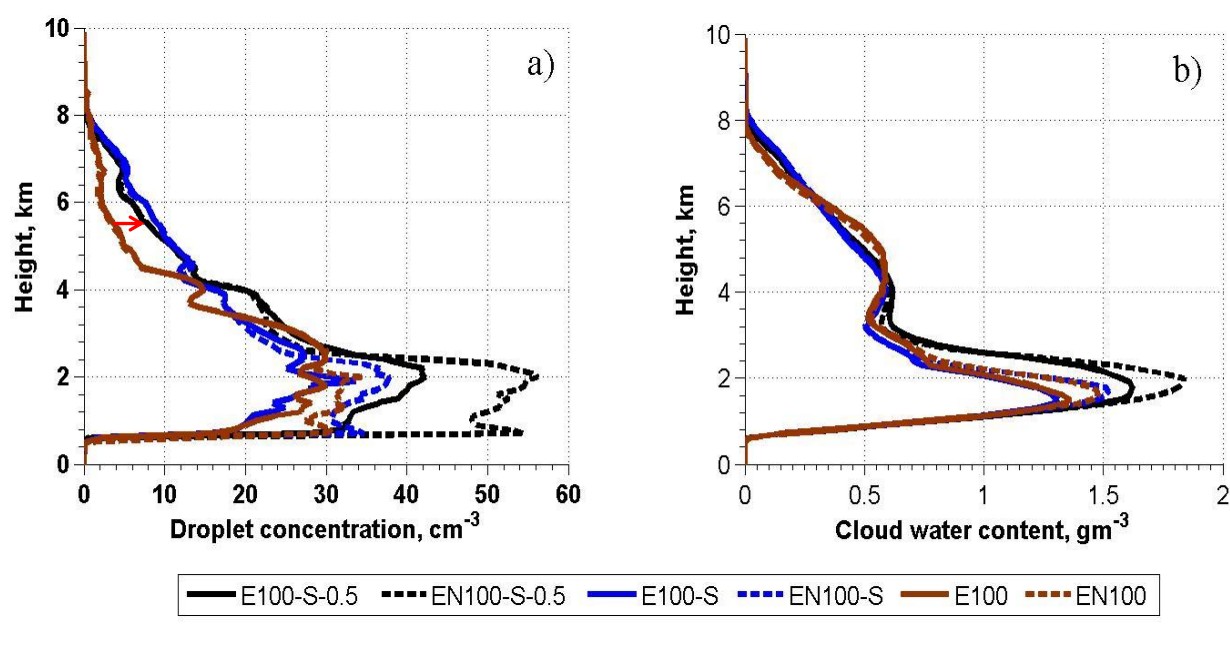



**Figure 8.** Vertical profiles of the maximum values of droplet concentration (a) and CWC (b) in simulations with low CCN concentration ($N_0 = 100 \ cm^{-3}$). The profiles are obtained by averaging over the time period of 3420-4020s. Red arrow shows the increase in droplet concentration due to in-cloud nucleation in simulations with the CCN spectra containing small CCN.





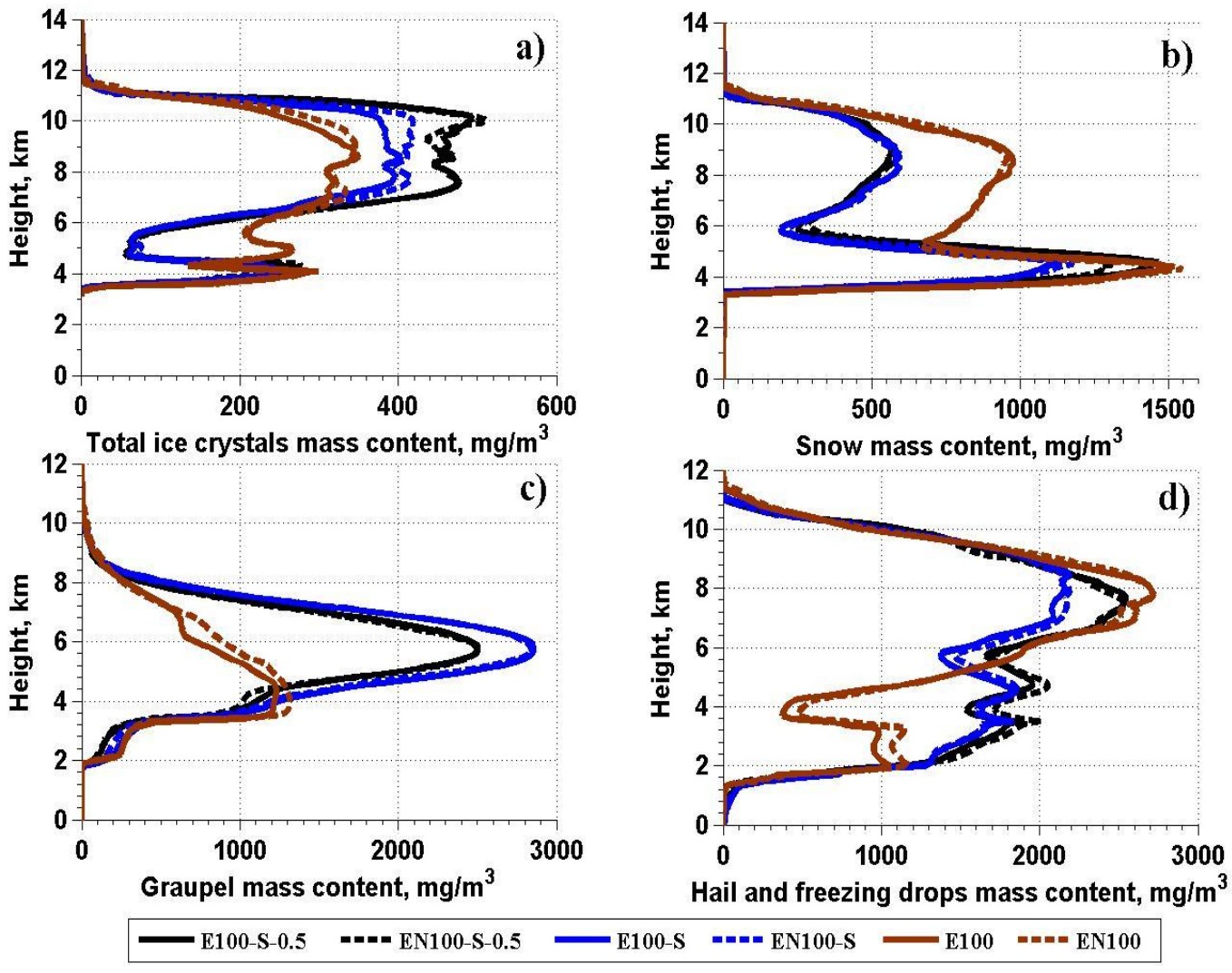

**Figure 9.** Vertical profiles of the maximum values of mass content: (a) total ice crystals, (b) snow, (c) graupel and (d) total hail and freezing drops in the simulations with low CCN concentration. The profiles are obtained by averaging over the time period of 3420-4020s.



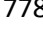

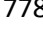




**Figure 10.** Time dependencies of (a) accumulated rain at surface for polluted and (b) for clean. Accumulated hail at the surface for polluted (c) and for clean (d) in different simulations in polluted cases.