# Peer review of "Application of a new scheme of cloud base droplet nucleation in a Spectral (bin) Microphysics cloud model: sensitivity to aerosol size distribution"

_Atmospheric Chemistry and Physics, 2016_

## Referee Comment (RC1) · Anonymous Referee #1 · 2 Aug 2016

Review on "Application of a new scheme of cloud base droplet nucleation in a Spectral (bin) Microphysics cloud model: sensitivity to aerosol concentrations" by Ilotoviz and Khain.

The results of numerical runs with new nucleation scheme are presented and compared to the standard scheme for which the supersaturation is calculated as a function of the mean thermodynamic variables of the grid box.

In the new scheme they estimate Smax using the method proposed by Pinsky et al. [2012], in which Smax is proportional to  $W^3/4$  and to Nd-1/2. They use this scheme

to correct for the underestimation that usually occurs in numerical models for which the grid size is too large to resolve Smax. They compare runs with and without the new scheme and show that it significantly improves the droplet concentration profiles, as compared with a parcel model, for polluted clouds.

This paper deals with an important problem of cloud resolving models. The resolution effect on the maximum supersaturation near cloud base is an important problem that potentially creates underestimation in the activation of the first mode of the droplets and offering an alternative approach to correct for this underestimation is an important task. The paper should be published but few clarifications will make it much clearer:

1) The paper relay heavily on the theoretical work presented in Pinsky 2012 and previous works. It would be nice to have this paper on a more "standalone mode". A summary of the main assumptions and derivations would make it much useful. 2) On the same note, throughout the paper the validation of the new scheme (NA) should be better explained. When the results are compared to a one D model - is it a parcel model? When the authors states that the results of the NA are "much better" they should explain more on how they reached this conclusion. 3) Does the model with the new scheme assigns Smax as the supersaturation for all of the gridbox near cloud base? If yes wouldn't it results in an overestimation of the activation? If not please explain why? 4) Is this parametrization done only for the gridbox near (above) cloud base? If yes how does the LCL is found? How sensitive is it to the location of the theoretical LCL within the gridbox? Say that in one case the theoretical LCL is toward the upper part of the gridbox, wouldn't it make more sense to assign the Smax parameter to the gridbox above? How sensitive it is to such details? 5) Smax and N (number of activated droplets) are coupled. Smax depends on N and N (or r(critical for activation)) on Smax. Could the authors explain how they solve them both and it the analytical parametrization? I guess one equation is eq. 3 but another equation is needed.

---

## Referee Comment (RC2) · Anonymous Referee #2 · 3 Aug 2016

The study is about testing a new scheme of droplet nucleation at cloud base using HUCM with spectral (bin) microphysics. The goals of the study are to test effects of the improved calculation of supersaturation maximum near cloud base (new approach-NA) at different aerosol loadings, and to evaluate sensitivity of cloud microphysics to concentration and shape of size distribution of aerosol particles. The goals are achieved generally but some conclusions need to be refined with additional investigation (see the comments below). The introduction is very simple can be expanded as well. The paper is well written, despite some grammar errors. Therefore, revisions are needed before being accepted by ACP.

[Figure]

Major comments, 1. The Introduction is short and sounds incomplete. It can be expanded to include (1) the importance of droplet nucleation to cloud properties and precipitation, (2) the description of the current approach (ST) and its limitation.

2. If the authors want to claim that the NA method gives more realistic droplet nucleation than ST and also to better evaluate both the ST and NA, it is necessary to conduct a benchmark test in which very high vertical resolution is used to resolve the maximum supersaturation and compare the supersaturation and droplet concentration with those from the benchmark test. In addition, the authors have a statement that the NA can be applied to any vertical resolution. By comparing Smax parameterized with the model predicted in such a test can help support that conclusion as well. This test should not be difficult to do with the idealized 2-D model.

3. Some clarification is needed for the description of NA method (Section 2) and additional discussion is needed throughout Section 4 (see the specific comments below).

Specific comments,

1) Line 134-139, for Eq (3), I am confused here, how did you solve three unknowns (Smax, the critical radius of the aerosols activated, and the nucleated droplet concentration) with Eq. 3? 2) Line 142, what is the new microphysical scheme? A little more details are needed here. 3) Line 147-151, do you mean the simulation is not initiated with a real sounding? Then I would like to see some justifications how the used dynamics and thermodynamics are close to a realistic atmosphere condition. 4) Line 161-162 and Line 167-168, for the clean conditions, why the minimum CCN radius is set to be a little smaller than the polluted conditions? 5) Line 196-200, Figure 2 does not show the results of Sw and droplet concentrations. Please present the results. 6) Line 253-256, the statement about "the decrease in the snow mass content" in the more droplet nucleation condition is not what Fig. 6 shows. Snow water content in EN3500-S is larger than E3500-S. Also, the NA method produces such greater graupel water content than the ST when shape parameter is 0.9. Is this related to a certain threshold used in the

riming processes to form grauple? In the tests with the shape factor of 0.5, the increase is not as dramatic. Why? 7) Figure 8 and Line 272-276, the discussion here should be compared with Figure 4 which shows that results for the high CCN condition. The differences between NA and ST in droplet number concentration are smaller, which is limited by available CCN. Also, CWC peaks at very different height compared with the high CCN condition. 8) Figure 9 and Line 297-298, the statement is not right about hail. The hail mass content is the largest in the E100 and EN100 where no smaller CCN exist and droplet concentration is the lower than others. In addition, please discuss such high sensitivity of graupel to the small CCN (i.e. droplet number concentration under the maritime cloud condition and give possible reasons about it. 9) Figure 10d and line 315-319, why does the hail precipitation in EN100-S-0.5 is much less than E100-S-0.5 since effect of small CCN is also included in this set of tests? 10) Line 345-347, I do not understand this statement, the small CCN increase droplet concentrations at the much higher levels, not around cloud case, how can it be made up by using the NA method? I did not see such results from Figures 4 and 8. I think the conclusion should be in-cloud nucleation has to be considered in the case of existing small CCN. 11) Line 366-368, see my comment in #8. 12) Line 382-383, the statement "It can be used in cloud-resolved models with different vertical grid spacing", is not supported by the content yet. By adding the benchmark test in which Smax calculated with NA can be compared with the model predicted Smax would address this problem.

―――――――――――――――――

---

## Author Comment (AC1) · 27 Sep 2016

We express our gratitude to Referee for the valuable comments and remarks.

Anonymous Referee #1

1. C. The paper relay heavily on the theoretical work presented in Pinsky 2012 and previous works. It would be nice to have this paper on a more "standalone mode". A summary of the main assumptions and derivations would make it much useful.

(R) We added more details in description of the approach in Section 2 (model de-
scription). A short derivation of the basic equation for supersaturation maximum is presented in new Appendix.

2. C. On the same note, throughout the paper the validation of the new scheme (NA) should be better explained. When the results are compared to a one D model – is it a parcel model? When the authors state that the results of the NA are "much better" they should explain more on how they reached this conclusion.

(R) We used the parcel model to calculate supersaturation maximum and concentration using CCN distribution and vertical velocity at the cloud base as in the HUCM simulations. New Fig 2 shows that New Approach produces the values of supersaturation and droplet concentrations much closer to "exact" values obtained by the parcel model than to the values obtained in ST.

3. (C) Does the model with the new scheme assigns Smax as the supersaturation for all of the gridbox near cloud base? If yes wouldn't it results in an overestimation of the activation? If not please explain why? (R) The values of Smax are calculated at all grid points that we assume to be associated to cloud base (the first grid point from below at which ). Some overestimation is possible in case of very high concentration as it is shown by Pinsky et al. (2012), but this error is substantially lower than in case Standard Approach is used.

4) (C) Is this parameterization done only for the gridbox near (above) cloud base? If yes how does the LCL is found? How sensitive is it to the location of the theoretical LCL within the gridbox? Say that in one case the theoretical LCL is toward the upper part of the gridbox, wouldn't it make more sense to assign the Smax parameter to the gridbox above? How sensitive it is to such details?

(R) We determine the model cloud base in the way described in the response above. In this approach, the grid point is slightly above the theoretical LCL, because we use condition . At the same time, the calculations performed according to Pinsky et al. (2012) show that the level where is located, i.e. from about 20 m (for high CCN concentration)

to about 60 m (for low CCM concentration) is higher than the LCL. The estimations show, therefore, that the level where is quite close to the model cloud base level. Accordingly, the droplet concentration determined at is assigned to the corresponding grid point at the model cloud base. We believe that the fact that we assign the droplet concentration calculated at the point of Smax to the lower model level, where ,does not lead to serious errors.

5) (C) Smax and N (number of activated droplets) are coupled. Smax depends on N and N (or r(critical for activation)) on Smax. Could the authors explain how they solve them both and it the analytical parametrization? I guess one equation is eq. 3 but another equation is needed.

(R) Detailed explanations are added in Section 2.

Please also note the supplement to this comment:
http://www.atmos-chem-phys-discuss.net/acp-2016-499/acp-2016-499-AC1-supplement.pdf

[Figure]

[Figure]

Vertical profile of supersaturation, W=1m/s, T=288K

Height above cloud base, m

E3500-S-0.5, $N_d$=416

EN3500-S-0.5, $N_d$=89

E100-S-0.5, $N_d$=31 cm

EN100-S-0.5, $N_d$=47

Parcel model: $N_d$=45

Parcel model: $N_d$=83

$S_{max}$=0.07%

$S_{max}$=0.34%

$S_{max}$=0.55%

$S_{max}$=1.

**Droplet concentration [cm³] (2400 sec)**

a)

[Figure]

**Droplet concentration [cm³] (2400 se**

b)

**Droplet concentration [cm³] (2400 sec)**

c)

[Figure]

**Droplet concentration [cm³] (2400 se**

d)

[Figure]

[Figure]

Vertical max, t=4860s-5460s a)

Height, km

Plates concentration, $l^{-1}$ $\times 10^5$

[Figure]

Averaged values

$\times 10^4$

b)

Plates concentration, $l^{-1}$

Time, s

[revised manuscript text omitted]

---

## Author Comment (AC2) · 27 Sep 2016

We express deep gratitude to Referee for the valuable comments and remarks. Anonymous Referee #2

1) The Introduction is short and sounds incomplete. It can be expanded to include (1) the importance of droplet nucleation to cloud properties and precipitation, (2) the description of the current approach (ST) and its limitation.

(R) The introduction is rewritten in more detail. It is stressed that droplet concentration determines major microphysical cloud properties such as height of precipitation onset, type of precipitation (liquid, mixed phase and ice). The description of Standard Approach is given, its limitations are mentioned. The necessity of exact calculation of cloud droplet concentration at cloud base is stressed. New Approach is described in more detail in Section 2. A new Appendix is added to describe the derivation of the basic equation for supersaturation maximum.

C. If the authors want to claim that the NA method gives more realistic droplet nucleation than ST and also to better evaluate both the ST and NA, it is necessary to conduct a benchmark test in which very high vertical resolution is used to resolve the maximum supersaturation and compare the supersaturation and droplet concentration with those from the benchmark test. In addition, the authors have a statement that the NA can be applied to any vertical resolution. By comparing Smax parameterized with the model predicted in such a test can help support that conclusion as well. This test should not be difficult to do with the idealized 2-D model.

(R) We conducted the benchmark test. The values of supersaturation and droplet concentrations in the vicinity of cloud base calculated using ST and NA were compared with supersaturation and droplet concentration calculated using a high precision parcel model. It is shown that results obtained using NA are much closer than ST results to those obtained by means of the parcel model.. This comparison is presented in new Fig. 2.

3.(C) Some clarification is needed for the description of NA method (Section 2)

(R) the description of NA is clarified. The derivation of the equations is presented in new Appendix.

(C) Additional discussion is needed throughout Section 4 (see the specific comments below). (R). The discussion is added (see responses to the specific questions).

Specific comments: 1)(C) Line 134-139, for Eq (3), I am confused here, how did you solve three unknowns (Smax, the critical radius of the aerosols activated, and the nucleated droplet concentration) with Eq. 3?

(R). The detailed explanation is presented in Section 2.

2) (C) Line 142, what is the new microphysical scheme? A little more details are needed here. (R) The new microphysical scheme is NA described in the model description section. The sentence was changed as follows: Effects of NA on cloud microphysics were tested...

3) C. Line 147-151, do you mean the simulation is not initiated with a real sounding? Then I would like to see some justifications how the used dynamics and thermodynamics are close to a realistic atmosphere condition.

(R) We use the surface temperature during the storm pass. To avoid confusion, the sentence was deleted.

4)(C) Line 161-162 and Line 167-168, for the clean conditions, why the minimum CCN radius is set to be a little smaller than the polluted conditions?

(R) According to Ghan et al. (2011, Table 2) the nuclei mode (the smallest CCN) in Marine aerosol size distribution contains aerosols smaller than the nuclei mode in Continental case or even than in Urban case. This comment is included into the revised paper.

5) (C) Line 196-200, Figure 2 does not show the results of Sw and droplet concentrations. Please present the results.

(R) New Fig. 2 is presented in the revised version.

6) C. Line 253-256, the statement about "the decrease in the snow mass content" in the more droplet nucleation condition is not what Fig. 6 shows. Snow water content in EN3500-S is lower than E3500-S. Also, the NA method produces such greater graupel water content than the ST when shape parameter is 0.9. Is this related to a certain threshold used in the C2 ACPD Interactive comment Printer-friendly version Discussion paper riming processes to form graupel? In the tests with the shape factor of 0.5, the increase is not as dramatic. Why?

(R) The text is rewritten in a clearer manner. Snow water mass content in EN3500-S is lower than E3500-S because rimming is more efficient in EN3500-S as compared to E3500-S (more supercooled water was nucleation at cloud base and ascent to higher levels increase the riming). The rimed snow is converted to graupel. This also explains the increase in mass content of graupel and hail in all NA cases. The difference in supercolled water is less pronounced when at slope parameter k=0.5 (see Fig. 4). The existence of the smallest CCN concentration (at k=0.9) leads to an increase in the differences between NA and ST. We attribute this difference to the fact that in E3500-S the liquid water content at upper levels is higher, which leads to larger graupel mass formed by riming.

7) C. Figure 8 and Line 272-276, the discussion here should be compared with Figure 4 which shows that results for the high CCN condition. The differences between NA and ST in droplet number concentration are smaller, which is limited by available CCN. Also, CWC peaks at very different height compared with the high CCN condition.

(R) The comparison is added. The differences between the cases plotted in Fig 4 and 8 are discussed in detail.

8)(C) Figure 9 and Line 297-298, the statement is not right about hail. The hail mass content is the largest in the E100 and EN100 where no smaller CCN exist and droplet concentration is the lower than others. In addition, please discuss such high sensitivity of graupel to the small CCN (i.e. droplet number concentration under the maritime cloud condition and give possible reasons about it.

(R) Done. The text is rewritten as follows:

The mass content of snow decreases with the increase in the smallest CCN concentration, because the smallest CCN increase supercooled drop content that leads to intensification of riming of snow. In turn, riming leads to its conversion to graupel (Fig. 9b).Consequently, the graupel mass content increases (Fig. 9c). As regards to mass content of hail, the increase in the the smallest CCN concentration leads to a decrease in the hail content above 6 km and to its increase below this level (Fig. 9d). The higher hail mass content above 6 km layer in the absence of smallest CCN is likely related to the fact the low droplet concentration leads to formation of raindrops in high concentration. Although these raindrops are of comparatively small size, the total raindrop mass content is larger than that in case of higher drop concentration. These raindrops rapidly freeze above the freezing level producing hail (actually frozen drops) with total mass larger than at high CCN concentration. This effect is discussed by Ilotovich et al. (2016) in detail. In HUCM, frozen raindrops are assigned to the hail category due to their high density. If hail is defined as particles with sizes exceeding 1 cm, the amount of hail at low CCN concentration would be negligible. Higher hail mass content below 6 km in the presence of the smallest CCN can be attributed to intense conversion of heavy rimed graupel to hail, as well as to more efficient hail growth by riming. In a deep convective cloud developing in the polluted atmosphere more hail particles from as compared to a cloud developing in clean air (Ilotovich et al. 2016). Due to larger size, hail in the polluted case falls to the surface (Fig. 6d), while in clean air hail melts at 1.5 km in the absence f no small CCN and in vicinity of the surface in case if the CCN size spectrum contains the smallest CCN.

9) C. Figure 10d and line 315-319, why does the hail precipitation in EN100-S-0.5 is much less than E100-S-0.5 since effect of small CCN is also included in this set of tests?

(R) Amount of hail at the surface in polluted air (Figure 10c) is substantially larger than in clean air s (Figure 10d) due to lower sizes and faster melting of hail particles at low CCN concentration. The effect of AP on the size and amount of hail at the surface was investigated by Ilotovich et al. (2016) in detail. The main factor determining the differences in the amount of hail falling to the surface in cases of low CCN concentration is effect of smallest CCN. The increase in concentration of smallest CCN leads to an increase in hail growth by riming.

As regards to the ratios of hail amounts in each group (with high and small CCN concentrations), for instance in the simulations EN100-S-0.5 and E100-S-0.5, these ratios change because of the earlier or later intensification of convective cells. Since the mass of hail falling to the surface (especially in clean air) is very low, a larger computational area is required to obtain reliable statistics and to get certain conclusions. Since at times exceeding about 200 min simulated cloud approached the lateral boundary, we re-plotted figure 10, so the time dependencies are shown till 180 min (instead of 220 min in the earlier version).

10)C. Line 345-347, I do not understand this statement, the small CCN increase droplet concentrations at the much higher levels, not around cloud base, how can it be made up by using the NA method? I did not see such results from Figures 4 and 8. I think the conclusion should be in-cloud nucleation has to be considered in the case of existing small CCN. (R) The error in the calculation of the droplet concentration near cloud base in ST is compensated to a significant extent by in-cloud nucleation above cloud base. Indeed, in NA droplet concentration increases with height up to the level of 4 km (Fig. 4a). The only reason of such increase is the in-cloud nucleation of comparatively large CCN. Smallest CCN are activated at higher levels. Corresponding comments are included into the revised paper. 11) C. Line 366-368, see my comment in #8. (R) The effect of the smallest CCN on hail is discussed in the body of the paper. The sentence pointed out by Referee is changed to: "The smallest CCN also influence hail size and mass content".

12)C. Line 382-383, the statement "It can be used in cloud-resolved models with different vertical grid spacing", is not supported by the content yet. By adding the benchmark test in which Smax calculated with NA can be compared with the model predicted Smax would address this problem.

(R). The comparison with the benchmark parcel model is included and discussed.

———————————————————

[revised manuscript text omitted]